# CATER: Intellectual Property Protection on Text Generation APIs via Conditional Watermarks

**Xuanli He**[*]
University College London
zodiac.he@gmail.com

**Qiongkai Xu** [*]
University of Melbourne
qiongkai.xu@unimelb.edu.au

**Yi Zeng**[†]
Virginia Tech
yizeng@vt.edu

**Lingjuan Lyu**[‡]
Sony AI
Lingjuan.Lv@sony.com

**Fangzhao Wu**
Microsoft Research Asia
fangzwu@microsoft.com

**Jiwei Li**
Shannon.AI, Zhejiang University
jiwei_li@shannonai.com

**Ruoxi Jia**
Virginia Tech
ruoxijia@vt.edu

## Abstract

Previous works have validated that text generation APIs can be stolen through imitation attacks, causing IP violations. In order to protect the IP of text generation APIs, recent work has introduced a watermarking algorithm and utilized the null-hypothesis test as a post-hoc ownership verification on the imitation models. However, we find that it is possible to detect those watermarks via sufficient statistics of the frequencies of candidate watermarking words. To address this drawback, in this paper, we propose a novel Conditional wATERmarking framework (CATER) for protecting the IP of text generation APIs. An optimization method is proposed to decide the watermarking rules that can minimize the distortion of overall word distributions while maximizing the change of conditional word selections. Theoretically, we prove that it is infeasible for even the savviest attacker (they know how CATER works) to reveal the used watermarks from a large pool of potential word pairs based on statistical inspection. Empirically, we observe that high-order conditions lead to an exponential growth of suspicious (unused) watermarks, making our crafted watermarks more stealthy. In addition, CATER can effectively identify IP infringement under architectural mismatch and cross-domain imitation attacks, with negligible impairments on the generation quality of victim APIs. We envision our work as a milestone for stealthily protecting the IP of text generation APIs.

## 1 Introduction

Nowadays, many technology corporations, such as Google, Amazon, Microsoft, have invested a plethora of workforce and computation to data collection and model training, in order to deploy well-trained commercial models as pay-as-you-use services on their cloud platforms. Therefore, these corporations own the intellectual property (IP) of their trained models. Unfortunately, previous works have validated that the functionality of a victim API can be stolen through imitation attacks, which

---

[*]Equal contribution. Most of the work was finished when X.H was at Monash Unversity.

[†]Work done during internship at Sony AI.

[‡]Corresponding author.

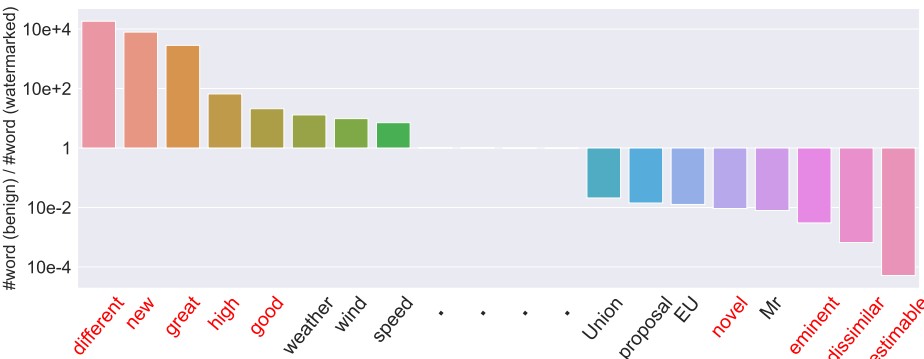

Figure 1: Ratio change of word frequency of top 100 words between benign and watermarked corpora used by [13], namely $P_b(w)/P_w(w)$. Red words are the selected watermarks. Although we only list 16 words having the most significant ratio change in the benign and watermarked corpora and omit the rest of them for better visualization, all watermarks are within the top 100 words.

inquire the victim with carefully designed queries and train an imitation model based on the outputs of the target API. Such attacks cause severe IP violations of the target API and stifle the creativity and motivation of our research community [44, 48, 20, 12, 9].

In fact, imitation attacks work not only on laboratory models, but also on commercial APIs [48, 51], since the enormous commercial benefit allures competing companies or individual users to extract or steal these successful APIs. For instance, some leading companies in NLP business have been caught imitating their competitors' models [40]. Beyond imitation attacks, the attacker could potentially surpass victims by conducting unsupervised domain adaptation and multi-victim ensemble [51].

In order to protect victim models, He et al. [13] first introduced a watermarking algorithm to text generation and utilized the null-hypothesis test as a post-hoc ownership verification on the imitation models. However, traditional watermarking methods generally distort the word distribution, which could be utilized by attackers to infer the watermarked words via sufficient statistics of the frequency change of candidate watermarking words. As an example shown in Figure 1, the replaced words and their substitutions are those with most frequency decrease ratios and increase ratios, respectively. To address this drawback, we are motivated to develop a more stealthy watermarking method to protect the IP of text generation APIs. The stealthiness of the new watermarks is achieved by incorporating high-order linguistic features as conditions that trigger corresponding watermarking rules.

Overall, our main contributions are as follows[4]:

- We propose a novel Conditional wATERmarking framework (CATER) for protecting text generation APIs. An optimization method is proposed to decide the watermarking rules that can *i)* minimize the distortion of overall word distributions, while *ii)* maximize the change of conditional word selections.
- Theoretically, we prove that a small number of the used watermarks could be blended in and camouflaged by a large number of suspicious watermarks when attackers attempt to inverse the backend watermarking rules.
- Empirically, we observe that high-order conditions lead to the exponential growth of suspicious (unused) watermarks, which encourages better stealthiness of the proposed defense with little hurt to the generation of victim models.

## 2 Preliminary and Background

### 2.1 Imitation Attack

An imitation attack (a.k.a model extraction) aims to emulate the behavior of the victim model $\mathcal{V}$, such that the adversary can either sidestep the service charges or launch a competitive service [44, 20, 48, 12, 51]. Malicious users can achieve this goal through interaction with the victim model $\mathcal{V}$ without knowing its internals, such as the model architecture, hyperparameters, training data, *etc.* Adversaries

---

[4]Code and data are available at: https://github.com/xlhex/cater_neurips.git

first craft a set of queries $Q$ based on the documentation of a target model. Then $Q$ will be sent to $\mathcal{V}$ to obtain the corresponding predictions $Y$. Finally, an imitation model $\mathcal{S}$ can be attained by learning a function to map $Q$ to $Y$.

Most prior imitation attacks are limited to classification tasks [44, 30, 12]. The imitation for text generation, a crucial task in natural language processing, has been under-developed until recently. Inspired by the efficacy of sequence-level knowledge distillation [17], Wallace et al. [48] and Xu et al. [51] propose mimicking the functionally of commercial text generation APIs. Similar to the standard imitation attack, adversaries can query $\mathcal{V}$ with $Q$. For generation tasks, $Y$ is a sequence of tokens $(y_1, ..., y_L)$, where $L$ is the length of the sequence. According to their empirical studies, one can rival the performance of these APIs, which poses a severe threat to cloud platforms.

## 2.2 Identification of IP Infringement

Prior works have utilized watermarking avenues to achieve a post-hoc verification of the ownership [45, 24, 25]. However, this line of work assumes the model owners can watermark victim model $\mathcal{V}$ by altering its neurons before releasing $\mathcal{V}$ to end-users. This operation is not feasible for the imitation attack, as $\mathcal{V}$ cannot access the parameters of $\mathcal{S}$. The only thing under the control of $\mathcal{V}$ is the responses to adversaries. Hence, some recent works propose creating a backdoor to $\mathcal{S}$ during the interaction with attackers [20, 42]. Specifically, $\mathcal{V}$ can select a small fraction of queries and answer them with incorrect predictions, in a similar way to the popular choice of watermarks in the computer vision domain (adopting some arbitrary features as the trigger to evaluate) [10, 54]. Erroneous predictions are so abrupt that $\mathcal{S}$ will memorize these outliers [6, 21]. As such, $\mathcal{V}$ can utilize these watermarks as evidence of ownership.

Albeit the efficacy, the drawbacks of backdoor approaches are tangible as well. First, since $\mathcal{V}$ does not impose regulations on users' usage, one cannot distinguish a malicious user from a regular user based on their querying behaviors[5]. Thus, $\mathcal{V}$ has to fairly serve all users and store all mislabeled queries, which leads to a massive storage consumption and a negative impact on the users' experiences. Moreover, as the identity of imitation models is unknown to $\mathcal{V}$, $\mathcal{V}$ has to iterate over all the mislabeled queries, which is computationally prohibitive. Finally, as $\mathcal{S}$ tends to adopt the pay-as-you-use policy for the sake of profits, the brute-force interaction with $\mathcal{S}$ can cause drastic financial costs.

As a remedy, He et al. [13] utilize a lexical watermark to identify IP infringement brought by imitation attacks. They point out that a neat watermarking algorithm must follow two principles: *i*) it cannot significantly impair customer experience, and *ii*) it should not be reverse-engineered by malicious users. In order to fulfill these requirements, they first select a set of words $\mathcal{W}$ from the training data of the victim model $\mathcal{V}$. Then for each $w \in \mathcal{W}$, they find $R-1$ semantically equivalent substitutions for it. Next, they employ $\mathcal{W}$ and their substitutions $\mathcal{T}$ to compose watermarking words $\mathcal{M}$. Finally, they replace $\mathcal{W}$ with $\mathcal{M}$. The rationale behind this avenue is to alter the distribution of words such that the imitation model can learn this biased pattern. To verify such a biased pattern of the word choice, He et al. [13] employ a null hypothesis test [36] for evaluation.

More concretely, He et al. [13] utilize an evaluation set $O$ to conduct the null hypothesis test. They formulate the null hypothesis as: *the tested model generates outputs without preference for watermarks*. A null hypothesis can be either rejected or accepted via the calculation of a p-value [36]. They assume that all words $\{w_i | w_i \in \mathcal{W} \cup \mathcal{T}\}$ follow a binomial distribution $Pr(k; n, p)$, where $k$ is the number of words in $\mathcal{M}$ appearing in $O$, $n$ is the number of words in $\mathcal{W} \cup \mathcal{T}$ found in $O$, and $p$ is the probability of watermarks observed in the natural language. According to their algorithm, $p$ is approximated by $1/R$. Now, one can compute the p-value from as follows:

$$\mathcal{P} = 2 \cdot \min(Pr(X \geq k), Pr(X \leq k)) \tag{1}$$

The p-value indicates how one can confidently reject the hypothesis. Lower p-value suggests that the tested model should be more likely subject to an imitator.

## 2.3 Watermark Removal

In conjunction with model watermarking, there is a growing body of investigations on watermark removal [45, 4, 53]. This line of work aims to erase watermarks embedded in white-box deep neural

---

[5]`https://cloud.google.com/translate/pricing`

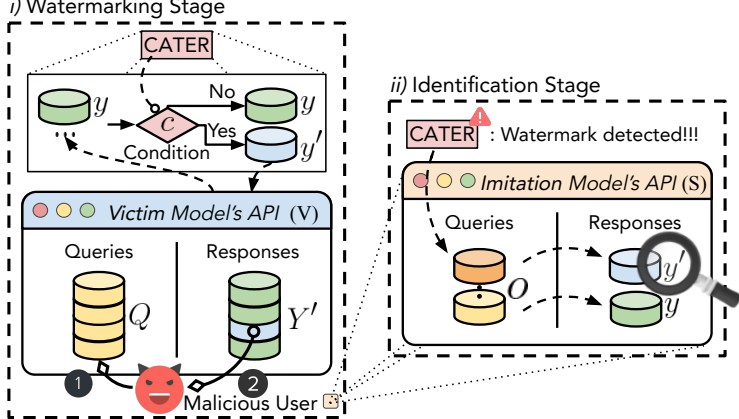

Figure 2: The workflow of CATER IP protection for Generation APIs. CATER first watermarks some of the responses from victim APIs (left). Then, CATER identifies suspicious attacker's API by watermark verification (right).

networks. We argue that these approaches are implausible for our setting, as text generation APIs are black-box to attackers.

Moreover, one can dub watermarking into a form of data poisoning [21, 50, 57, 49], in which one can utilize trigger words to manipulate the behavior of the victim model. A list of works has investigated how to mitigate the adverse effect caused by data poisoning in NLP tasks. Qi et al. [33] show that GPT2 can effectively identify trigger words targeting the corruption of text classifications. It has been demonstrated that one can use influence graphs as a means of the remedy for data poisoning on various NLP tasks [41]

## 3 CATER

This section introduces our proposed *CATER*, leveraging conditional watermarks to watermark the imitation model, which can be served as a post-hoc identification of an IP infringement. Figure 2 provides an overview of CATER, consisting of two stages as below.

*i)* **Watermarking Stage:** The victim API model $\mathcal{V}$ employs CATER to add conditional watermarks to the intended responses. When the vanilla victim model receives queries $Q = \{q_i\}_{i=1}^{|Q|}$ from an end-user, $\mathcal{V}$ initially produces a tentative answer $Y = \{y_i\}_{i=1}^{|Q|}$. Next, $\mathcal{V}$ utilizes CATER to conduct a watermarking procedure over $Y$ according to the watermarking rules, given condition $c$. Finally, $\mathcal{V}$ replies to the end-user with a watermarked response $Y'\{y_i'\}_{i=1}^{|Q|}$.

*ii)* **Identification Stage:** If a model $\mathcal{S}$ is under suspicion, the victims can query the suspect using a verification set $O = \{o_i\}_{i=1}^{|O|}$. After obtaining the responses $Y = \{y_i\}_{i=1}^{|O|}$ from $\mathcal{S}$, $\mathcal{V}$ can leverage CATER to testify whether $\mathcal{S}$ violates the IP right of $\mathcal{V}$.

### 3.1 Watermarking Rule Optimization

Watermarking some words to a deterministic substitutions could distort the overall word distribution. Therefore, some watermarks could be reversely inferred and eliminated by analyzing the word distribution, as demonstrated in Figure 1. We propose to inject the watermarks in conditional word distribution, while maintaining the original word distribution. The substitutions can be conditioned on linguistic features as illustrated in Figure 3. Remarkably, given a condition $c \in \mathcal{C}$ and a group of semantically equivalent words $\mathcal{W}$, one can replace any words $w \in \mathcal{W}$ with each other. We formulate the objective of conditional watermarking rules as:

$$\min_{\hat{P}(w|c)} \underbrace{\mathbb{D}\big(\sum_{c \in \mathcal{C}} \hat{P}(w|c)P(c), \sum_{c \in \mathcal{C}} P(w|c)P(c)\big)}_{\text{I: indistinguishable objective}} - \frac{\alpha}{|\mathcal{C}|} \underbrace{\sum_{c \in \mathcal{C}} \mathbb{D}\big(\hat{P}(w|c), P(w|c)\big)}_{\text{II: distinct objective}} \qquad (2)$$

The two factors reflect two essential desiderata:

***i)*** For each $\mathcal{W}$, with $w \in \mathcal{W}$, the overall word distributions before optimization $P(w) = \sum P(w|c)P(c)$ and after optimization $\hat{P}(w) = \sum \hat{P}(w|c)P(c)$ should be close to each other, as the *indistinguishable objective* in Equation 2;

***ii)*** For a particular condition $c \in \mathcal{C}$, the conditional word distributions should still be distinct to their original distributions, reflected by the dissimilarity between $P(w^{(i)}|c)$ and $\hat{P}(w^{(i)}|c)$, as the *distinct objective* in Equation 2. This guarantees the conditional watermarks are identifiable in verification. In practice, we utilize multiple synonym word sets as a group $\mathcal{G} = \{\mathcal{W}^{(i)}\}_{i=1}^{|\mathcal{G}|}$. For each $\mathcal{W}^{(i)}$, we can formulate Equation 2 as a mixed integer quadratic programming using $\ell_2$-norm as distance measurement function:

$$\min_{\boldsymbol{W}} (\boldsymbol{Wc} - \boldsymbol{Xc})^T (\boldsymbol{Wc} - \boldsymbol{Xc}) - \frac{\alpha}{|\mathcal{C}|} \text{Tr}((\boldsymbol{W} - \boldsymbol{X})^T (\boldsymbol{W} - \boldsymbol{X}))$$

$$\text{s.t. } \boldsymbol{X}^T \cdot \boldsymbol{1}_{|\mathcal{W}^{(i)}|} = \boldsymbol{1}_{|\mathcal{C}|}, \boldsymbol{X} \in \{0,1\}^{|\mathcal{W}^{(i)}| \times |\mathcal{C}|} \tag{3}$$

We define matrix $\boldsymbol{X} = [\hat{P}(w^{(i)}|c)]_{|\mathcal{W}^{(i)}| \times |\mathcal{C}|}$ as the variables for optimization. Matrix $\boldsymbol{W} = [P(w^{(i)}|c)]_{|\mathcal{W}^{(i)}| \times |\mathcal{C}|}$ and vector $\boldsymbol{c} = [P(c)]_{|\mathcal{C}| \times 1}$ are constant variables, decided by calculating corresponding distributions in a large training corpus. The objective of Equation 3 is convex when $\alpha$ is sufficiently small (see the proof in Appendix A). We optimize the watermark assignments $\hat{P}(w^{(i)}|c)$ using Gurobi [11] with $\alpha = 0.01$.

### 3.2 Constructing Watermarking Conditions using Linguistic Features

This part will concentrate on practical ways to construct the watermarking conditions $\mathcal{C}$. We consider two fundamental linguistic features $\mathcal{F}$, *i)* part-of-speech and *ii)* dependency tree, and their high-order variations as conditions. Such linguistic features were widely and successfully used for text classification [5, 52], sequence labeling [22, 37], and *etc.*

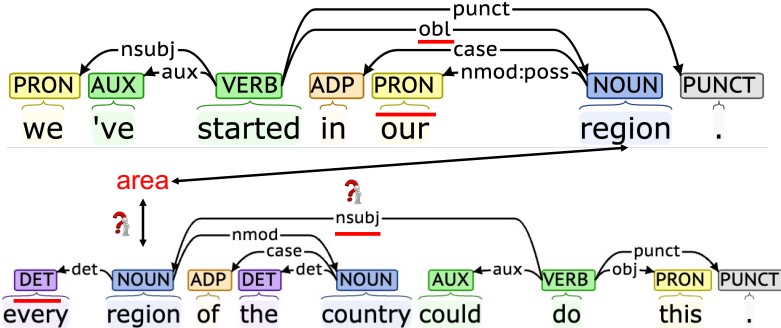

Figure 3: The part-of-speech (POS) tags and dependency relations are illustrated in colored boxes and arcs, respectively. The decision of using "region" or its synonym ("area" or other words), is conditioned on its linguistic features in context. For example, in the first sentence, either "PRON" (POS label of the preceding token) or "obl" (DEP label of the incoming arc) gives the decision to replace "region" with "area", as opposed to "DET" or "nsubj" in the second sentence.

**Part-of-Speech** Part-of-Speech (POS) tagging is a grammatical grouping algorithm, which can cluster words according to their grammatical properties, such as syntactic and morphological behaviors [16]. The POS tag for each token is demonstrated in a colored box, in Figure 3.

Given a word $w$ in a sentence, we denote its POS as $l_0$, and use $l_{-k}$ or $l_{+k}$ to represent the POS of the $k$-th word to the left or right of $w$. We consider a single label $l_{-1}$ as our first-order condition. In order to reduce the identifiability of our conditional watermark, we can construct high-order conditions from the same feature set, *e.g.*, $(l_{-1}, l_{+1})$ as second-order condition and $(l_{-2}, l_{-1}, l_{+1})$ as third-order condition. Note that if $l_{-k}$ or $l_{+k}$ does not exist, we use a pseudo tag "[none]" by default. Since POS describes grammatical roles of words and its classes are limited, the combination of POS of an

anchor and its neighbors should also be bounded. Thus one can consider the POS bond among words as the condition.

**Dependency Tree**    Dependency Tree (DEP) is a syntactic structure, which describes directed binary grammatical relations between words [16], as shown in Figure 3. A dependency tree can be represented by an acyclic directed graph $G = (V, E)$, where $V$ is a set of vertices corresponding to all words in a given sentence, $E$ is a set of ordered pairs of vertices, denoted as *arcs*. An arc $e \in E$ describes a grammatical relation between two vertices in $V$, *i.e.,* source vertex named as *head* and target vertex coined as *dependent*. Except the root vertex, each vertex is connected to by exactly one head. Consequently, there exists a unique path from each vertex to the root node in a dependency tree.

Analagously for POS features, we can design first-order and high-order DEP features as watermarking conditions. Given a word $w$, and its incoming DEP arc, we use the DEP label of the arc as the first-order features ($d_1$). We construct high-order condition recursively using the labels of incoming DEP arcs ($d_1, d_2, \cdots$). A pseudo arc label "[none]" is used when there is no parent node in recursion.

### 3.3    Identifiability of Conditional Watermark

In this section, we discuss the identifiability of our watermark if the attackers attempt to infer the used watermarks. We assume the worst case that the attackers have access to *i)* the watermarking algorithm, *ii)* all possible word sets for substitution $\mathcal{G}$, and *iii)* combination of feature sets $\mathcal{F}$ as wartermarking conditions $\mathcal{C}$. However, the exact watermarking rules are unobservable to attackers. The attackers may identify the watermark rules by suspecting those observed $P(w^{(i)}|c)$ with extreme distributions, *i.e.*, only a single word in a synonym set is selected given a specific condition.

Given a limited budget, we assume that an imitator has queried our watermarked API and has acquired $N$ tokens in $\bigcup_i \mathcal{W}^{(i)}$. The system has incorporated watermarks with $K$-order features $c \in \mathcal{C}$, where $\mathcal{C} = (\mathcal{F}_1, \mathcal{F}_2, \cdots \mathcal{F}_K)$, the total number of possible conditions is $|\mathcal{C}| = \prod_{i=1}^{K} |\mathcal{F}_i|$. We simplify our discussion by using the same feature set $\mathcal{F}$ (POS or DEP), then $|\mathcal{C}| = |\mathcal{F}|^K$.

**Theorem 3.1.** *If $|\mathcal{F}|^K > N$ and there exist $t$ conditions that have less or equal to $m \in \mathbb{Z}^+$ support samples, then $t \geq |\mathcal{F}|^K - N/(m+1)$.*

The attacker would suspect conditional word distributions that are *extremely imbalanced*, namely only a single dominant choice of word is observed within the responses to the attacker.

**Theorem 3.2.** *Having $m$ support samples for a specific condition $c$, the possibility of observing extremely imbalanced word choice is $\mathcal{I}(\mathcal{W}, c, m) = \sum_{w_i \in \mathcal{W}} P(w_i|c)^m$. If $m' \leq m$ and $m, m' \in \mathbb{Z}^+$, $\mathcal{I}(\mathcal{W}, c, m') \geq \mathcal{I}(\mathcal{W}, c, m)$.*

The proofs of Thm 3.1 and Thm 3.2 can be found in Appendix B and C. Thm 3.1 guarantees a lower bound for the number of conditions that attackers will have less or equal to $m$ observed samples. Moreover, the lower bound grows exponentially with regard to the feature orders used as watermarking conditions. Thm 3.2 guarantees the high probability of the conditions with extremely imbalanced word selection when $m$ is small. Combining these two theorems, the total number of the suspicious watermark rules could be huge compared with the limited used watermark rules if we are utilizing high-order linguistic features as conditions. We further empirically demonstrate the significant confusion between suspected and used watermark rules in Section 4.2.

## 4    Experiments

**Text Generation Tasks.**    We examine two widespread text generation tasks: machine translation and document summarization, which have been successfully deployed as commercial APIs.[6][7]. To demonstrate the generality of CATER, we also apply it to two more text generation tasks: *i)* **text simplification** and *ii)* **paraphrase generation**. We present the performance of CATER for these tasks in Appendix F.4.

- **Machine Translation:** We consider WMT14 German (De) →English (En) translation [2] as the testbed. We follow the official split: train (4.5M) / dev (3,000) / test (3,003). Moses [18] is

---

[6]https://translate.google.com/

[7]https://deepai.org/machine-learning-model/summarization

Table 1: Performance of different watermarking approaches on WMT14 and CNN/DM. We use F1 scores of ROUGE-1, ROUGE-2 and ROUGE-L for CNN/DM.

| | WMT14 | | | CNN/DM | | |
| --- | --- | --- | --- | --- | --- | --- |
| | p-value ↓ | BLEU ↑ | BERTScore ↑ | p-value ↓ | ROUGE-1/2/L ↑ | BERTScore ↑ |
| w/o watermark | $> 10^{-1}$ | 31.1 | 65.9 | $> 10^{-1}$ | 37.7 / 15.4 / 31.2 | 22.1 |
| Venugopal et al. [47] | | | | | | |
|   - unigram | $< 10^{-2}$ | 30.4 | 65.2 | $< 10^{-2}$ | 37.3 / 15.1 / 31.2 | 21.7 |
|   - trigram | $> 10^{-1}$ | 30.8 | 65.7 | $> 10^{-1}$ | 37.5 / 15.3 / 31.0 | 21.8 |
|   - sentence | $> 10^{-1}$ | 30.8 | 65.9 | $> 10^{-1}$ | 37.6 / 15.4 / 31.2 | 21.9 |
| He et al. [13] | | | | | | |
|   - spelling | $< 10^{-13}$ | 31.1 | 65.8 | $< 10^{-8}$ | 37.5 / 15.2 / 31.4 | 22.0 |
|   - synonym | $< 10^{-10}$ | 30.8 | 65.5 | $< 10^{-8}$ | 37.6 / 15.3 / 31.4 | 21.8 |
| CATER (ours) | | | | | | |
|   - DEP | $< 10^{-4}$ | 30.9 | 65.4 | $< 10^{-2}$ | 37.6 / 15.3 / 31.3 | 21.8 |
|   - POS | $< 10^{-7}$ | 30.8 | 65.3 | $< 10^{-7}$ | 37.5 / 15.2 / 31.2 | 21.9 |

applied to pre-process all corpora, with a cased tokenizer. We use BLEU [32] and BERTScore [55] to evaluate the translation quality. BLEU concentrates on lexical similarity via n-grams match, whereas BERTScore targets at semantic equivalence through contextualized embeddings.

- **Document summarization:** CNN/DM [14] utilizes informative headlines as summaries of news articles. We reuse the dataset preprocessed by See et al. [38] with a partition of train/dev/test as 287K / 13K / 11K. Rouge [26] and BERTScore [55] are employed for the evaluation metric of the summary quality.

We use 32K and 16K BPE vocabulary [39] for experiments on WMT14 and CNN/DM, respectively.

**Models.** For the primary experiments, we consider Transformer-base [46] as the backbone of both victim models and the imitation models. Following He et al. [13], we use a 3-layer Transformer for the summarization task. Because of their superior performance, pre-trained language models (PLMs) have been deployed on cloud platforms.[8] Hence, we also consider using two popular PLMs: *i)* BART (summarization) [23] and *ii)* mBART (translation) [27] as the victim model. Regarding the imitation model, since the architecture of the victim model is unknown to the adversary, we simulate this black-box setting by using three different architectures as the imitator, namely (m)BART, Transformer-base, and ConvS2S [8]. The training details are summarized in Appendix D.

**Basic Settings.** As a proof-of-concept, we start our evaluation with a most straightforward case. We assume the victim model $\mathcal{V}$ and the imitation model $\mathcal{S}$ use the same training data, but $\mathcal{S}$ uses the response $y'$ with CATER instead of the ground-truth $y$. We set the size of synonyms to 2 and vary this value in Appendix F.1. The detailed construction of watermarks and approximation of $p$ in Equation 1 for CATER is provided in Appendix D.

**Baselines.** We compare our approach with [47] and [13]. Venugopal et al.[47] proposed watermarking the generated output with a sequence of bits under the representation of either n-grams or the complete sentence. He et al. [13] devises two effective watermarking approaches. The first one replaces all the watermarked words with their synonyms. The second one watermarks the victim API outputs by mixing American and British spelling systems.

## 4.1 Performance of CATER

Table 1 presents the watermark identifiability and generation quality of studied text generation tasks. Both [13] and CATER obtain a sizeable gap in the p-value, and demonstrate a negligible degradation in BLEU, ROUGE, and BERTScore, compared to the non-watermarking baseline. However, [47] falls short of injecting detectable watermarks. Although CATER is slightly inferior to [13] in p-value, we argue that watermarks in [13] can be easily erased, as their replacement techniques are not invisible. As shown in Figure 1, the synonyms used by [13] can be identified due to the tangible distribution shift on the watermarks, whereas CATER manages to minimize such a shift according to Equation 2, which is also corroborated by Figure 7. In addition, one can eliminate the spelling watermarks by consistently using one spelling system.

---

[8]https://cloud.google.com/ai-platform/training/docs/algorithms/bert

Table 2: Imitation performance of different architectures on clean and watermarked data. Numbers in parentheses are results of clean data. Victim models are trained on mBART (WMT14) and BART (CNN/DM), respectively. We use the first-order POS as the watermarking approach.

| Model | WMT14 | | CNN/DM | |
|---|---|---|---|---|
| | p-value ↓ | BLEU ↑ | p-value ↓ | ROUGE-L ↑ |
| (m)BART | $< 10^{-4} (> 10^{-1})$ | 34.9 (35.2) | $< 10^{-5} (> 10^{-1})$ | 38.1 (38.1) |
| Transformer | $< 10^{-5} (> 10^{-2})$ | 32.7 (33.0) | $< 10^{-3} (> 10^{-1})$ | 32.8 (32.9) |
| ConvS2S | $< 10^{-5} (> 10^{-2})$ | 32.7 (32.9) | $< 10^{-3} (> 10^{-1})$ | 32.7 (32.7) |

Figure 4: BLEU scores (**left**) and p-value (**right**) of using different orders of the POS watermarking approach on WMT14 data. X-axis indicates the orders of conditions. *1*, *2*, *3* represent the first-order, second-order, and third-order condition respectively. **Clean** means imitation with clean dataset.

Unless otherwise stated, we use the first-order POS as the default setting for CATER, due to its efficacy in terms of watermark identifiability and generation quality.

**IP Identification under Architectural Mismatch**   The architectures of remote APIs are usually unknown to the adversary. However, recent works have shown that the imitation attack is effective even if there is an architectural mismatch between the victim model and the imitator [48, 12]. To demonstrate that our approach is model-agnostic, we use BART-family models as victim models and vary architectures of imitation models.

Table 2 summarizes p-value and generation quality of CATER on WMT14 and CNN/DM datasets. Similar to Table 1, CATER can confidently identify the IP infringement when the architecture of the imitation model is the same as that of the victim model, with a gap of p-value between watermarked model and benign model being 3 orders of magnitude. In addition, this gap applies to the case, where we use distinct architectures for the victim model and the imitator. The generation quality exhibits negligible drops, within a range of 0.3. Note that the generation quality of Transformer and ConvS2S imitators degrades due to the capacity gap, compared to powerful BART-family models.

**IP Identification on Cross-domain Imitation**   Similarly, the training data of the victim model is confidential and remains unknown to the public. Thus, there could be a domain mismatch between the training data of the victim model and queries from the adversary. In order to exhibit that our approach is exempt from the domain shift, we use two out-of-domain datasets to conduct the imitation attack for the machine translation task. The first is IWSLT14 data [3] with 250K German

Table 3: Imitation performance of using data from different domains. The victim model is trained on WMT14. We use first-order POS as the watermarking condition.

| WMT14 | IWSLT14 | OPUS (Law) |
|---|---|---|
| $< 10^{-7}$ | $< 10^{-5}$ | $< 10^{-6}$ |

sentences, and the second is OPUS (Law) data [43] consisting of 2.1M German sentences. Table 3 suggests that despite the domain mismatch, CATER can still watermark the imitation model, and one can identify watermarks with high confidence.

**High-order Conditions**   We have shown that the first-order CATER effectively performs various tasks and settings. We argue that CATER is not limited to the first-order condition. Instead, one can use high-order CATER as mentioned in Section 3.2, which can consolidate the invisibility as discussed in Section 3.3. Therefore, we investigate the efficacy of the high-order CATER to the translation task and provide the study on the summarization task in Appendix F.2.

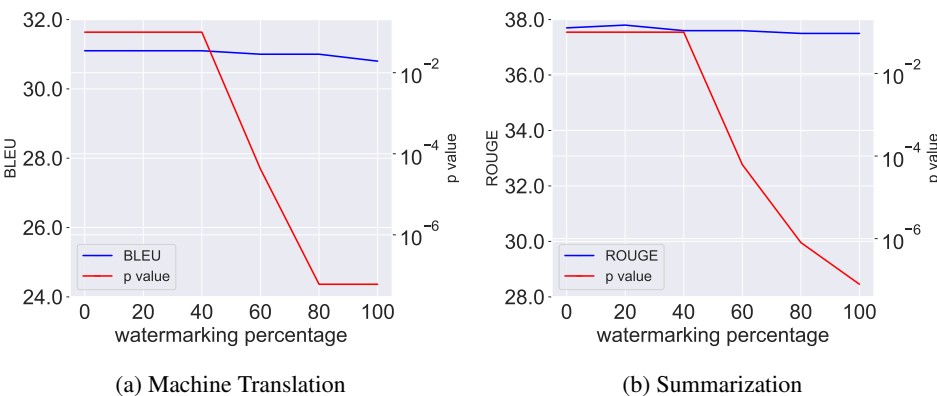

|  | | |
| (a) Machine Translation | | (b) Summarization |

Figure 5: The generation quality and p-value under different percentage of using watermarked data for imitation attacks on machine translation and summarization.

Table 4: Imitation performance with watermark removal on WMT14 data and CNN/DM. We use the first-order POS as the watermarking approach. ONION is used to remove watermarks.

| Model | WMT14 | | CNN/DM | |
|---|---|---|---|---|
| | p-value $\downarrow$ | BLEU $\uparrow$ | p-value $\downarrow$ | ROUGE-L $\uparrow$ |
| w/o ONION | $< 10^{-7}$ | 30.8 | $< 10^{-7}$ | 31.2 |
| w/ ONION | $< 10^{-5}$ | 27.0 | $< 10^{-7}$ | 25.9 |

Figure 4 shows that with the increase of conditional POS orders, compared to the use of clean data, there is no side effect on the BLEU scores, *i.e.,* generation quality. The right figure suggests that using the higher conditional orders can lead to the larger p-value, which means that the claim about IP violation is less confident. However, the gap between the benign and watermarked models is still significantly distinguishable. Note that for the sake of fair comparison, the p-values of the clean model are calculated *w.r.t.* the corresponding order.

**Mixture of Human- and Machine-labeled Data** Due to multiple factors, such as noisy inputs [19, 1], domain mismatch [1, 29], *etc.,* training a model with machine translation alone still underperforms using human-annotated data [51]. However, since annotating data is resource-expensive [51], malicious may mix the human-annotated data with machine-annotated one. We examine the effectiveness of CATER under this mixture of two types of datasets.

Figure 5 suggests that as CATER aims to minimize the distribution distortion, watermarks injected by CATER tend to be overwritten by clean signals. Thus, CATER is active when more than half of the data is watermarked.

## 4.2 Analysis on Adaptive Attacks

The previous sections illustrates the efficacy of CATER for watermarking and detecting potential imitation attacks. Given the case that a savvy attacker might be aware of the existence of watermarks, they might launch countermeasures to remove the effects of the watermark. This section explores and analyzes possible adaptive attacks based on varying degrees of prior knowledge of our defensive strategy. Specifically, we examine two types of adaptive attacks that try to erase the effects of the watermark: *i)* **vanilla watermark removal**, and *ii)* **watermarking algorithm leakage**.

**Vanilla Watermark Removal.** Under this setting, we assume the attackers are aware of the existence of watermarks, but not aware of the details of the watermarking algorithm. Following such a setting of attacker knowledge, we assume the attacker would adopt an existing watermark removal technique in their vanilla form. To evaluate, we employ ONION, a popular defensive avenue for data poisoning in the natural language processing field, which adopts GPT-2 [35] to expel outlier words. The defense results are shown in Table 4. We find that ONION cannot erase the injected watermarks. Meanwhile, it drastically diminishes the generation quality of the imitation model.

**Watermarking Algorithm Leakage.** Under this case study, we assume attackers have access to the full details of our watermarking algorithm, *i.e.,* the same watermarking dictionary and the features for constructing watermarking conditions. We note that this is the most substantial attacker knowledge assumption we can imagine, aside from the infeasible case that they know the complete pairs of watermarks we used.

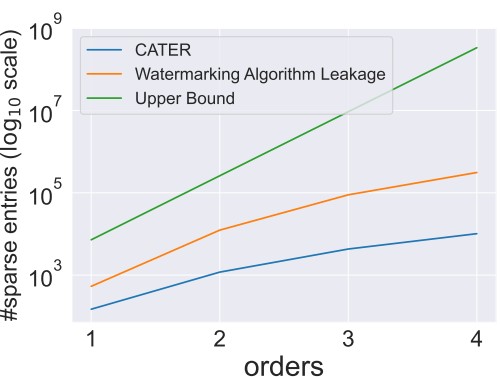

After collecting responses from the victim model, the attackers can leverage the leaked knowledge to analyze the responses to find the used watermarks, *i.e.,* the number of sparse entries. As shown in Section 3.3, we theoretically prove that such reverse engineering is infeasible. In addition, Figure 6 shows that even with such a strong attacker knowledge, the amount of potential candidate watermarks (orange curve) is still astronomical times larger than the used number of watermarks (blue curve). Thus, malicious users would have difficulty removing watermarks from the responses; unless they lean toward modifying all potential watermarks. Such a brute-force approach can drastically debilitate the performance of the imitation attack, causing a feeble imitation. Finally, we demonstrate the upper bound (green curve) to show that without the curated knowledge about the watermarking conditions, the attackers have to consider all possible combinations of POS tags. Therefore, the difficulty of identifying the watermarks from the top 200 words can be combinatorially exacerbated.

Figure 6: The number of sparse entries (suspected watermarks) of top 200 words with watermarking algorithm leakage under different orders (orange) on the training data. CATER indicates the actual number of watermarks used by our watermarking system (blue). POS feature is used where $|\mathcal{F}| = 36$. The upper bound indicates all possible combinational watermarks (green).

## 5 Conclusion

In this work, we are keen on protecting text generation APIs. We first discover that it is possible to detect previously proposed watermarks via sufficient statistics of the frequencies of candidate watermarking words. We then propose a novel Conditional wATERmarking framework (CATER), for which, an optimization method is proposed to decide the watermarking rules that can minimize the distortion of overall word distributions while maximizing the change of conditional word selections. Theoretically, we prove that it is infeasible for even the savviest attackers, who know how CATER algorithms, to reveal the used watermarks from a large pool of potential watermarking rules based on statistical inspection. Empirically, we observe that high-order conditions lead to an exponential growth of suspicious (unused) watermarks, rendering our crafted watermarks more stealthy.

### Limitation and Negative Societal Impacts

One major limitation of our work is that one has to find high-quality synonym sets to minimize semantic degradation, leading to the limited option of candidate words. Nevertheless, according to Section 4, given the top 200 words and their synonyms, CATER can still achieve a stealthy watermarking. In addition, because of the use of the lexical match, we experience slight performance degradation in generation quality. Furthermore, since defending against imitation attacks is difficult, we resort to a post-hoc verification. If the adversaries do not publically release the imitation model, CATER becomes fruitless.

Regarding the negative societal impacts, CATER might be overused by some APIs owners as a means of unfair competition. As shown in Figure 4, the gap between the benign model and the watermarked one is small. Hence, the APIs owners could leverage CATER to sue innocent cloud services. As a remedy, we suggest the judges refer to a relatively higher bar, *e.g.,* lower p-value $< 10^{-6}$.

### Acknowledgments and Disclosure of Funding

This research was funded by Sony AI. We would like to appreciate the valuable feedback from all anonymous reviewers.

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
