# A Proof of the object in Equation 3 is convex, when $\alpha$ is sufficiently small.

To validate this statement, we first prove two factors in the object are convex (Lemma A.1 and Lemma A.2) and the combination of them keeps the convex property (Lemma A.3).

**Lemma A.1.** *The quadratic term $M_1 = (Wc - Xc)^T(Wc - Xc)$ is convex.*

*Proof.* We conduct first-order and second-order derivative of $M_1$ on $X$:

$$\frac{\partial M_1}{\partial X} = 2(Xc - Wc)c^T \tag{4}$$

$$\frac{\partial^2 M_1}{\partial X^2} = 2 \cdot (c \cdot c^T) \otimes I_{|\mathcal{C}|} \tag{5}$$

where $\otimes$ is Kronecker product and $I_{|\mathcal{C}|}$ is an identity matrix. Because $c \neq 0$, $c \cdot c^T$ is positive semidefinite, *i.e.*, $\text{Tr}(c \cdot c^T) \geq 0$,

$$\text{Tr}\Big(\frac{\partial^2 M_1}{\partial X^2}\Big) = 2 \cdot \text{Tr}(c \cdot c^T) \cdot \text{Tr}(I_{|\mathcal{C}|}) \geq 0.$$

Therefore, $\frac{\partial^2 M_1}{\partial X^2}$ is positive semidefinite and $M_1$ is convex. $\qquad\square$

**Lemma A.2.** *The quadratic term $M_2 = \text{Tr}((X - W)^T(X - W))$ is convex.*

*Proof.* We conduct first-order and second-order derivative of $M_2$ on $X$:

$$\frac{\partial M_2}{\partial X} = 2X - 2W \tag{6}$$

$$\frac{\partial^2 M_2}{\partial X^2} = 2 \cdot I_{|\mathcal{C}|} \otimes I_{|\mathcal{C}|} \tag{7}$$

Similar to the proof of Lemma A.1, we have

$$\text{Tr}\Big(\frac{\partial^2 M_2}{\partial X^2}\Big) = 2 \cdot \text{Tr}(I_{|\mathcal{C}|}) \cdot \text{Tr}(I_{|\mathcal{C}|}) \geq 0.$$

Therefore, $\frac{\partial^2 M_2}{\partial X^2}$ is positive semidefinite and $M_2$ is convex. $\qquad\square$

**Lemma A.3.** *Given two positive semidefinite matrices $P, Q \in \mathbb{R}^{N \times N}$, and a constant $0 \leq \alpha \leq \frac{\lambda_{\min}(P)}{\lambda_{\max}(Q)}$, $P - \alpha Q$ is positive semidefinite.*[9]

*Proof.* $P$ and $Q$ are positive semidefinite indicates that $\forall i \in [\![1..N]\!]$,

$$0 \leq \lambda_{\min}(P) \leq \lambda_i(P) \leq \lambda_{\max}(P), \tag{8}$$
$$0 \leq \lambda_{\min}(Q) \leq \lambda_i(Q) \leq \lambda_{\max}(Q). \tag{9}$$

Then, we have

$$\text{Tr}\big(P - \alpha Q\big) = \text{Tr}\big(P\big) - \alpha \text{Tr}\big(Q\big) \tag{10}$$

$$= \sum_{i=1}^{N} \lambda_i(P) - \alpha \sum_{i=1}^{N} \lambda_i(Q) \tag{11}$$

$$\geq N \cdot \big(\lambda_{\min}(P) - \alpha \lambda_{\max}(Q)\big) \geq 0. \tag{12}$$

Additionally, since $P$ and $Q$ are symmetric, $P - \alpha Q$ is also symmetric. Thus, $P - \alpha Q$ is positive semidefinite. $\qquad\square$

Combining Lemma A.1, Lemma A.2 and Lemma A.3, the objective of Equation 3 is convex when $\alpha$ is small.

---

[9] $\lambda_i(\cdot)$ is the $i$-th eigenvalue of a matrix. $\lambda_{\max}(\cdot)$ and $\lambda_{\min}(\cdot)$ respectively represent the maximum and minimum eigenvalues of a matrix.

## B Proof of Theorem 3.1.

*Proof.* There are $t$ conditions that have $[0, m]$ support samples, then the other $|\mathcal{F}|^K - t$ conditions should have $[m + 1, +\infty)$ support samples. Combining these two cases, we have

$$t \cdot 0 + (|\mathcal{F}|^K - t) \cdot (m + 1) \leq N, \tag{13}$$

therefore,

$$t \geq |\mathcal{F}|^K - N/(m + 1).$$

$\square$

## C Proof of Theorem 3.2.

We assume the observation of the triggered watermark words are independent to each other, as those words are sparsely distributed in our corpus (4 per 1000 words).

*Proof.* We prove $\mathcal{I}(\mathcal{W}, c, m') \geq \mathcal{I}(\mathcal{W}, c, m)$ by recursion: $\mathcal{I}(\mathcal{W}, c, m + 1) \geq \mathcal{I}(\mathcal{W}, c, m)$.

$$\mathcal{I}(\mathcal{W}, c, m + 1) = \sum_{w_i \in \mathcal{W}} P(w_i|c)^{m+1} \tag{14}$$

$$= \sum_{w_i \in \mathcal{W}} \left( P(w_i|c)^m \cdot P(w_i|c) \right) \tag{15}$$

$$\leq \left( \sum_{w_i \in \mathcal{W}} P(w_i|c)^m \right) \cdot \left( \sum_{w_i \in \mathcal{W}} P(w_i|c) \right) \tag{16}$$

$$= \sum_{w_i \in \mathcal{W}} P(w_i|c)^m = \mathcal{I}(\mathcal{W}, c, m). \tag{17}$$

$\square$

Note that a special case of Thm 3.2 is that $\mathcal{I}(\mathcal{W}, c, m) = \sum_{w_i \in \mathcal{W}} P(w_i|c) = 1$, when $m = 1$.

## D Additional Experimental Setup

**Construction of Watermarks**    To build watermarks from two semantically equivalent words, we use top 200 frequent words from the training set as the candidate words. For each word, we use WordNet [7] to find its synonyms and build a list of word sets. We notice that word sets include words that are not strictly semantically equivalent. Thus we use a pre-trained Word2Vec [28] to filter out sets with dissimilar words. In addition, to avoid replacement clash, we do not allow any word to appear in more than word set. Eventually, top 50 semantically matching pairs are retained for CATER. For linguistic features, we construct conditions from the POS tags and dependency trees, which are produced by Stanza [34]. We use Equation 3 to obtain preliminary watermarks $\{X_i\}_{i=1}^{50}$. We sort $\{X_i\}_{i=1}^{50}$ in ascending order according to the indistinguishable objective in Equation 2 and choose the top 10 of them as effective watermarks for CATER.

**Training Details**    We use fairseq [31] as our codebase. For the experiments of Transformer base, we train both victim and imitation models for 50 epochs. Following [46], we set the learning rate to 0.0005 with warmup steps of 4,000. We use a batch of 4,096 tokens per GPU. Then, we decrease the learning rate proportionally to inverse square root of the step number. We follow the training setup used in [23] and [27] for BART and mBART. All experiments are conducted on an NVIDIA DGX node with 8 V100 GPUs.

**Estimation of Watermarking Probability for CATER**    Given a group of semantically equivalent words $\mathcal{W}^{(i)}$ and the corresponding condition $c$, we denote $w_c^{(i)}$ as a basic unit, which depicts $w^{(i)}$ under the condition of $c$. If the conditional post-watermark distribution $\hat{P}(w^{(i)}|c)$ is 1 according to our algorithm, we consider $w_c^{(i)}$ as a watermark. Now, given a set of groups $\mathcal{G} = \{\mathcal{W}^{(i)}\}_{i=1}^{|\mathcal{G}|}$, we can find all watermarks and denote them as $\mathcal{M}$. We use $\#(\mathcal{M}, \mathcal{D}_{tr})$ to represent the count of words in $\mathcal{M}$

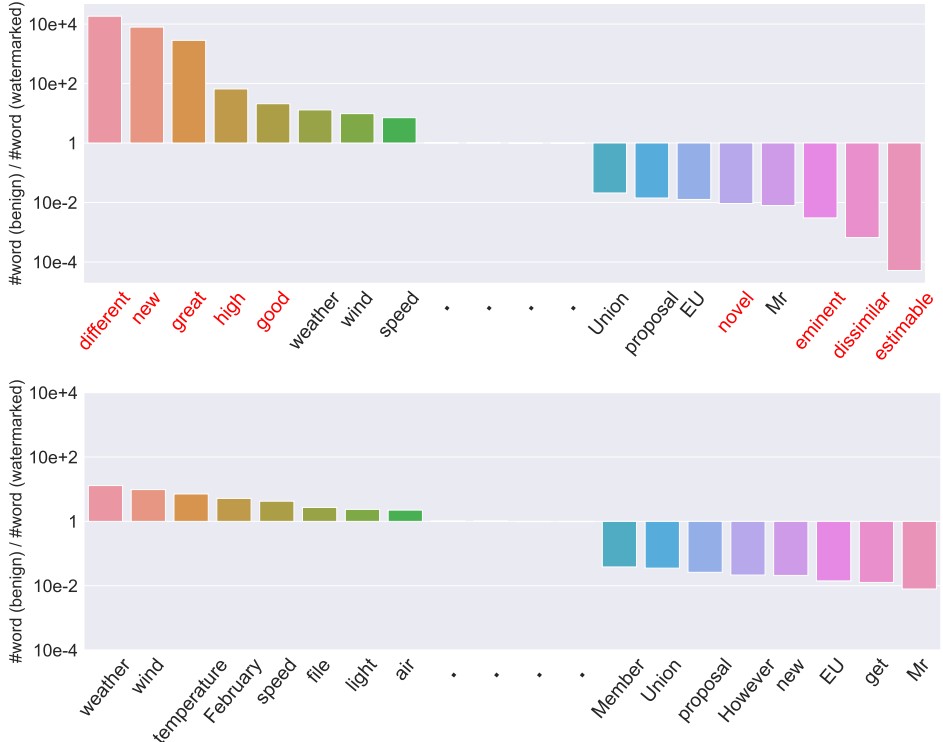

Figure 7: Ratio change of word frequency of top 100 words between benign and watermarked corpora, namely $P_b(w)/P_w(w)$. **Top**: watermarks are built from [13]. Red words are the selected watermarks. Although we only list 16 words having the most significant ratio change in the benign and watermarked corpora and omit the rest of them for better visualization, all watermarks are within the top 100 words. **Bottom**: watermarks are built from CATER. There is no significant ratio change for watermarked words.

appeared in the training data $\mathcal{D}_{tr}$ of the victim model. Similarly, we denote the count of all candidate words in $\cup_i \mathcal{W}^{(i)}$ as $\#(\cup_i \mathcal{W}^{(i)}, \mathcal{D}_{tr})$. Finally, the approximated $p$ in Equation 1 for CATER can be computed as:

$$p = \frac{\#(\mathcal{M}, \mathcal{D}_{tr})}{\#(\cup_i \mathcal{W}^{(i)}, \mathcal{D}_{tr})}$$

## E  Word Distribution Shift on Watermarked Response

To demonstrate that the watermarking algorithm of He et al. [13] can cause a drastic change in word distribution, whereas CATER is able to retain the word distribution, we compare the difference between the watermarked data with a clean one. Since the training data of the victim model is unknown to the malicious users, we randomly select 5M sentences from common crawl data as the benign corpus. Then we obtain the word distribution for the watermarked and benign corpora, respectively, denoted as $P_w$ and $P_b$. Next, we take the union of top 100 words of both watermarked and benign corpora to obtain the suspicious word set $S$. Finally, we can calculate the ratio change of word frequency of each word in $S$ between benign and watermarked corpora, namely $P_b(w)/P_w(w)$.

As shown in Figure 7, the watermarks injected by [13] can be easily identified, because of the sizeable word distribution shift. Instead, our approach manages to disguise the watermarks, which leads to more stealthy protection as expected.

## F Ablation Study

### F.1 Performance of CATER using Different Sizes of Synonyms

Section 4.1 shows that using two semantically equivalent words can effectively protect the IP right of the victim model. According to Section 3.1, CATER can be scaled to multiple semantically equivalent words. Our preliminary experiments show that finding more than 4 interchangeable words is not easy. Thus, we set $|\mathcal{W}^{(i)}|$ to 2,3 and 4. Table 5 shows that with $|\mathcal{W}^{(i)}|$ increased, CATER becomes more confident in identifying the IP infringement, which is observed in [13] as well. We attribute this phenomenon to the decline of $p$, *i.e.,* the chance of hitting the watermarks. Particularly, using more semantically equivalent words means that $p$ can decrease in normal data. Accordingly, the p-value of the watermarked model will drastically drop based on Equation 1.

Table 5: Watermarking performance of different sizes of synonyms. Numbers in parentheses are results of clean data. We use the first-order POS as the watermarking approach.

| $|\mathcal{W}^{(i)}|$ | WMT14 | |
| :---: | :---: | :---: |
| | **p-value** $\downarrow$ | **BLEU** $\uparrow$ |
| 2 | $< 10^{-7}$ $(> 10^{-2})$ | 30.8 (31.1) |
| 3 | $< 10^{-10}$ $(> 10^{-1})$ | 30.9 (31.1) |
| 4 | $< 10^{-14}$ $(> 10^{-1})$ | 30.9 (31.1) |

### F.2 High-order CATER for Summarization

This section provides the performance of CATER with high-order conditions on summarization tasks. Similarly, according to Figure 8, the high-order conditions do not have negative impact on the generation quality. Despite the increase in p-value, the high-order CATER is capable of watermarking the imitation model.

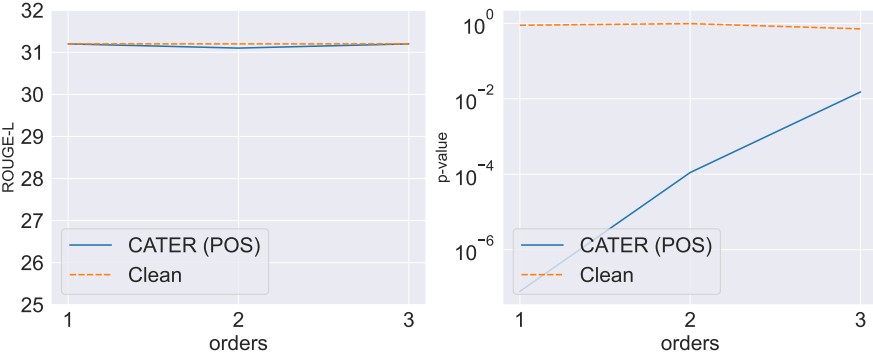

Figure 8: ROUGE-L scores and p-value of using different orders of the POS watermarking approach on CNN/DM data.

### F.3 Importance of Distinct Objective

We have discussed the importance of the distinct objective in Section 3.1. To validate our claim, we conduct an ablation study by removing this factor from our optimization objective. Table 6 suggests that we can easily distinguish a watermarked model with distinct object with a benign one, while the watermarked models without distinct objective possess almost the same performance as benign ones $(> 10^{-2})$, which corroborates our argument in Section 3.1.

### F.4 CATER for More Text Generation Tasks

To examine the generality of our approach, we run two additional generation tasks: *i)* text simplification and *ii)* paraphrase generation. We use wiki-large data [56] for text simplification, and QQP

Table 6: Watermarking performance of with (w/) and without (w/o) distinctive objective on WMT14 data. Numbers in parentheses are results of clean data. We use the POS conditions as the watermarking approach.

| Orders | w/ distinctive | | w/o distinctive | |
|---|---|---|---|---|
| | p-value ↓ | BLEU ↑ | p-value ↓ | BLEU ↑ |
| 1 | $< 10^{-7}$ ($> 10^{-2}$) | 30.8 (31.1) | $> 10^{-2}$ ($> 10^{-2}$) | 30.9 (31.1) |
| 2 | $< 10^{-6}$ ($> 10^{-2}$) | 30.9 (31.1) | $> 10^{-2}$ ($> 10^{-2}$) | 31.0 (31.1) |
| 3 | $< 10^{-3}$ ($> 10^{-2}$) | 30.9 (31.1) | $> 10^{-2}$ ($> 10^{-2}$) | 31.0 (31.1) |

Table 7: Performance of CATER on text simplification and paraphrase.

| | Text Simplification | | | Paraphrase | | |
|---|---|---|---|---|---|---|
| | p-value ↓ | SARI ↑ | BERTScore ↑ | p-value ↓ | BLEU ↑ | BERTScore ↑ |
| w/o watermark | $> 10^{-1}$ | 37.1 | 72.5 | $> 10^{-1}$ | 32.1 | 72.4 |
| CATER (POS) | $< 10^{-6}$ | 37.0 | 71.6 | $< 10^{-6}$ | 32.1 | 72.4 |

data[10] is used for paraphrase generation [15]. Similar to machine translation, we use Transformer base as the backbone. According to Table 7, CATER is effective on those tasks as well. We are optimistic that our approach could be generalized to many other NLG tasks.

# G   Watermarked Samples

| |
|---|
| Example 1: (area->region)
    original:        I ask the Commission : what can be done to speed up implementation in this particular area ?
    watermarked: I ask the Commission : what can be done to speed up implementation in this particular region ? |
| Example 2: (information->data)
    original:        There are various things that can undermine consumer confidence , for example the lack of information .
    watermarked: There are various things that can undermine consumer confidence , for example the lack of data . |
| Example 3: (help->aid)
    original:        All this will be managed with the help of modern technology in any area of the European Union .
    watermarked: All this will be managed with the aid of modern technology in any area of the European Union . |
| Example 4: (responsibility->obligation)
    original:        We have to remember that we share responsibility in that region with the international community .
    watermarked: We have to remember that we share obligation in that region with the international community . |
| Example 5: (film->movie)
    original:        The whole situation remembers an horror film . It just scares .
    watermarked: The whole situation remembers an horror movie . It just scares . |

Table 8: Watermarked samples with different watermarks.

---

[10]https://www.kaggle.com/c/quora-question-pairs