# OpenReview forum: "CATER: Intellectual Property Protection on Text Generation APIs via Conditional Watermarks"
_NeurIPS.cc/2022/Conference — NeurIPS 2022 Accept_

### Official Review · Reviewer_24GZ · 2022-07-03

**Rating:** 6
**Confidence:** 3
**Soundness:** 3 good
**Presentation:** 2 fair
**Contribution:** 3 good

**Summary:**

First of all, thank you for sharing great work with the community.

With the great success of language generative models, many models are employed in a business. So, the research community focused on the watermarking method to protect these models. The authors pointed out the vulnerability of existing watermarking methods against imitation attacks. They showed that existing works could be breakable when sufficient statistics are available. Therefore, this paper proposed a new framework named Conditional wATERmarking (CATER). The CATER framework is based on linguistic features (e.g., part-of-speech and dependency tree).

**Questions:**

1. Is the framework also working for other text generation tasks? (Connected with cons. in the above section.)
2. Is there a possibility of a collusion attack?
For example, there are two victim models using CATER for watermarking.
And the adversary trains his/her model using both two victim models.
In this case, who is the victim between both two victims?


**Limitations:**

As I mentioned above, the paper needs more empirical experiments on different text generation tasks.
And also, the figures need to be updated for the quality of the paper.
Based on the discussion period, the score could be updated.

---
Thank you for the response.
The authors resolve the questions I have.
And the additional experimental results are promising.
Therefore, I decided to increase my current score.

**Strengths And Weaknesses:**

Strengths
1. The authors gave mathematical proof of their framework's identifiability.
2. CATER is robust against white-box attacks.

Weaknesses
1. Experiments are limited. I hope the authors also need to provide another generation task (e.g., Inference, Chat).
---
<Minor suggestions to improve the quality>

1. Fig2. needs to be updated. The flow of the figure is not well matched with explanations.
2. Please bold the best score in Table 1,2.
3. Fig 4. needs more explanations.

---

> ### Author Response · Authors · 2022-08-02
> **Response to 24GZ**
>
> We would like to thank you for your insightful suggestions and encouraging feedback. We hope the additional experiments and our clarifications can consolidate our work.
>
> ---
> **Q1:** Is the framework also working for other text generation tasks?
>
> **A1:** Thanks for the suggestion. To examine the generality of our approach by running two additional generation tasks: 1) text simplification and 2) paraphrase generation. We use wiki-large data [1] for text simplification, and QQP data [2] is used for paraphrase generation. Similar to machine translation, we use Transformer base as the backbone. Following [1, 2], we use SARI and BLEU to evaluate the generation quality of text simplification and paraphrase generation, respectively.
>
> | [Text Simplification] | p-value   $\downarrow$ | SARI $\uparrow$ | BERTScore $\uparrow$  | [Paraphrase] |  p-value $\downarrow$   | BLUE $\uparrow$ | BERTScore $\uparrow$  |
> | :---        | :----:  |   :----:   |   :---: |               :---: | :----:  |   :----:   |          :---: |
> | w/o watermarking   | > $10^{-1}$ | 37.1      | 72.5   | w/o watermarking | > $10^{-1}$ | 32.1 |72.4 |
> | CATER   | **< $10^{-6}$** | 37.0      | 71.6      | CATER  | **< $10^{-6}$**  | 32.1 | 72.4|
>
> As shown in the table above, CATER is effective on those tasks as well. We are optimistic that our approach could be generalized to many other NLG tasks, as our approach is conducted on general languages.
>
> We hope these results have addressed your concerns.
>
> ---
> **Q2:** Update fig 2 to match the explanation
>
> **A2:** Thanks for the suggestion. We have updated Figure 2 and the corresponding description accordingly.
>
> ---
> **Q3:** Elaboration on Figure 4
>
> **A3:** The x-axis indicates the orders of the POS condition. As described in **section 3.2**, the first order means the POS of the left word of the target word, the second order means the POSs of the left and right words of the target word, and the third order means the POSs of the second left, left and right words of the target word. The orange line indicates an imitation model using the clean response from the victim model, whereas the blue line represents the watermarked imitation model.
> - The left figure shows that with the increase in the condition orders, the BLEU scores (or the generation quality) do not suffer a performance drop.
>
> - The right figure suggests that with the increase in the condition orders, the p-values of watermarked models become larger, which means that the claim about IP violation is less confident. However, the watermarked models are still detectable compared to the clean model. Please note that for the sake of fair comparison, the p-values of the clean model are calculated w.r.t the corresponding order. We have updated the description in our revised version between line 288 and line 293 as well.
>
> ---
> **Q4:** Can CATER address collusion attacks? For example, there are two victim models using CATER for watermarking.
>
> **A4:** Thanks for the suggestion. For the case that the attacker imitates multiple victim models protected by CATER, we believe that the collusion is very rare in the real-world setting due to:
> 1.  CATER uses a very tiny number of words for watermarks. In our experiments, only $3.8*10^{-5}$ of the words are watermarked. Thus, it is unlikely that multiple victims share the same watermarks.
> 2. As shown in **section 4.2**, the number of conditions could grow to infinite.
>
> To conclude, victim models can claim their ownership properly without confusing each other.
>
> We hope this has addressed your concern about collusion attacks.
>
> ----
> [1] Sentence Simplification with Deep Reinforcement Learning. Zhang et al. EMNLP 2017
>
> [2] Hierarchical Sketch Induction for Paraphrase Generation.  Hosking et al. ACL 2022

---

> > ### Author Response · Authors · 2022-08-07
> > **Thanks to Reviewer 24GZ**
> >
> > We would like to thank the reviewer for taking the time to review our paper and the valuable feedback, and in particular for admitting our work with a valuable research direction, good motivation, and sound theoretical proof.
> >
> > We hope our response has adequately addressed your concerns regarding more experiments on other text generation tasks, adding more details in figures, and discussing collusion attacks.
> >
> > We are more than happy to respond to any further questions if you still have concerns. We truly appreciate your valuable feedback and comments that help us further improve our work.

---

### Official Review · Reviewer_eyPy · 2022-07-10

**Rating:** 8
**Confidence:** 5
**Soundness:** 4 excellent
**Presentation:** 4 excellent
**Contribution:** 4 excellent

**Summary:**

This paper presents a simple but effective approach (called CATER) to protect the IP of text generation APIs under imitation attacks. This idea can stamp watermarks on imitation models by altering the distribution of semantically equivalent words. Unlike the previous work, they propose to conditionally watermark the victim’s outputs so that it is infeasible for the adversary to crack watermarks over the watermarked response, both theoretically and empirically.

The authors employ two linguistical rules as the conditions. The first one utilizes the part-of-speech tags of surrounding words as the condition. The second condition is established on the incoming arc of the dependency tree. According to their empirical studies, CATER can effectively watermark imitation models and identity the watermarks under various settings. In addition, the authors have shown that CATER is resilient to two strong watermark removal approaches.


**Questions:**

1.	Since APIs aim to serve end-users, in addition to automatic metrics, would it be possible to conduct a human evaluation before concluding that the watermarks do not affect generation quality?

2.	Would you please elaborate on the calculation of the p-value on CATER? It is unclear to me.


**Limitations:**

As far as I know, ONION focused on insertion-based removal. However, CATER utilizes substitution-based watermarking. So, probably the authors could try a substitution-based removal technique.

**Strengths And Weaknesses:**

Strengths:

1.	The paper found that analyzing the word distribution could reveal watermarks used by the previous watermarking approach (He et al.). Hence, the authors devise a conditional watermarking algorithm as a remedy.

2.	To perform the conditional watermarking and minimize the distribution shift observed in He et al., the authors formulate these two objectives as a linear programming problem. In addition, they prove that this optimization problem is solvable.

3.	The authors also rigorously prove that watermarks injected by CATER are infeasible to be inferred by statistical reverse-engineering, especially when scaling to high-order features.

4.	The paper shows that CATER rivals previous watermarking approaches and is effective in various settings, such as cross-domain querying and architectural mismatch.

Overall, I think this is a quite good paper, which may bring groundbreaking impact on protecting IP of NLP models.

Weakness:

The paper is of good quality and easy to follow. I have no major concerns but a few comments as below.

1.	Since APIs aim to serve end-users, in addition to automatic metrics, it would be good to conduct a human evaluation before concluding that the watermarks do not affect generation quality.

2.	Would you please elaborate on the calculation of the p-value on CATER? It is unclear to me.

---

> ### Author Response · Authors · 2022-08-02
> **Response to eyPy**
>
> We would like to thank you for your valuable and encouraging feedback. We consolidate our work by i) adding an emergency human evaluation and ii) more explanation to our paper.
>
> ---
> **Q1:** would it be possible to conduct a human evaluation before concluding that the watermarks do not affect generation quality?
>
> **A1:** Thanks for the valuable comments. Due to the short window for the rebuttal, we sample 100 instances to inspect their semantics before and after watermarking and score them with a 5-point Likert scale (1: strongly disagree, 5: strongly agree). The average score of these instances is 3.9. Thus, we believe the watermarks have minor effects on generation quality. We also demonstrate several examples to show the minor modification by CATER.
>
> Example 1: (area->region)
>
> - *I ask the Commission : what can be done to speed up implementation in this particular area ?*
>
> - *I ask the Commission : what can be done to speed up implementation in this particular region ?*
>
> Example 2: (information->data)
>
> - *There are various things that can undermine consumer confidence , for example the lack of information .*
>
> - *There are various things that can undermine consumer confidence , for example the lack of data .*
>
> Finally, we have asked external annotators to evaluate the data quality, and we will include their evaluation in our paper soon.
>
> ---
> **Q2:** elaborate on the calculation of p-value on CATER
>
> **A2:** Given a group of semantically equivalent words $\mathcal{W}^{(i)}$ and the corresponding condition $c$, we denote $w_{c}^{(i)}$ as a basic unit, which depicts $w^{(i)}$ under the condition of $c$. If the conditional post-watermark distribution $\hat{P}(w^{(i)}|c)$ is $1$ according to our algorithm, we consider $w_{c}^{(i)}$ as a watermark. Now, given a set of groups $\mathcal{G}=\\{\mathcal{W}^{(i)}\\}_{i=1}^{|\mathcal{G}|}$,
>
> we can find all watermarks and denote them as $\mathcal{M}$. We use $\\#(\mathcal{M}, \mathcal D_{tr})$ to represent the count of words in $\mathcal{M}$ appeared in the training data $\mathcal D_{tr}$ of the victim model. Similarly, we denote the count of all candidate words in $\cup_i \mathcal{W}^{(i)}$ as $\\#(\cup_i \mathcal{W}^{(i)}, \mathcal D_{tr})$. Finally, the approximated $p$ in Equation~1 for CATER can be computed as: $$p=\frac{\\#(\mathcal{M}, \mathcal D_{tr})}{\\#(\cup_i \mathcal{W}^{(i)}, \mathcal D_{tr})}$$
> Then, we use Equ 1 (line 103) to calculate the p-value with $p$.
>
> We have modified the corresponding descriptions in Section 3.1 and Appendix D.
>
> We hope the description will clarify the calculation for you.
>
> ---
> **Q3:** probably the authors could try a substitution-based removal technique.
>
> **A3:**  Thanks for the suggestion. To the best of our knowledge, there is no existing study for removing substitution-based data poisoning. Thus, we resort to insertion-based removal. We agree that this would be an interesting study, and we will leave it to our future work, when we find appropriate off-the-shelf substitution-based removal tools.

---

### Official Review · Reviewer_hUhu · 2022-07-10

**Rating:** 8
**Confidence:** 5
**Soundness:** 3 good
**Presentation:** 4 excellent
**Contribution:** 4 excellent

**Summary:**

This paper proposes a watermarking technique to claim ownership of text generation APIs when subject to imitation attacks. They show that prior work manages to watermark and detect imitation behaviors, but a weakness exists. After analyzing the word frequency in API responses and publicly available data, the adversary can identify the watermarks with a higher chance. Based on this observation, they devise a novel method CATER, which can conditionally watermark the response so that the adversary cannot decipher the watermarking keys. Meanwhile, CATER can identify the watermarked imitators as effectively as the previous work.

CATER leverages two linguistic features to fulfill the conditional watermarking: 1) neighboring part-of-speech tags and 2) dependency relations. They work well on two popular text generation tasks. They show that CATER can scale to high-order settings, which are empirically challenging to be identified by attackers. According to their experiments, the proposed watermarks have negligible adverse impacts on the utility.


**Questions:**

Please refer to the weakness part.

**Limitations:**

The authors only study CATER for the English-centric datasets. However, as we know, the widespread text generation APIs are for translation, which supports multiple languages. Probably, the authors could extend CATER to other languages in the future.

**Strengths And Weaknesses:**

Strength:
1.	Most works in protecting IP from imitation attacks focus on classification tasks. Wallace et al. (2020) have shown that imitation attacks are effective on commercial translation APIs. However, as an urgent vulnerability, little work has been done to protect the IP of text generation APIs, except He et al. (2021). This work identifies and addresses a major weakness in He et al., and proposes a more invisible watermarking algorithm, making their method more appealing to the community.
2.	Instead of using a heuristic search, the authors elegantly cast the watermark search issue into an optimization problem and provide rigorous proof.
3.	The authors conduct comprehensive experiments to validate the efficacy of CATER in various settings, including an architectural mismatch between the victim and the imitation model and cross-domain imitation.
4.	This work theoretically proves that CATER is resilient to statistical reverse-engineering, which is also verified by their experiments.  In addition, they show that CATER can defend against ONION, an effective approach for backdoor removal.

Weakness:
1.	The authors assume that all training data are from the API response, but what if the adversary only uses the part of the API response?
2.	Figure 5 is hard to comprehend. I would like to see more details about the two baselines presented in Figure 5.

---

> ### Author Response · Authors · 2022-08-02
> **Response to Reviewer hUhu**
>
> We would like to thank you for your valuable and encouraging feedback. We consolidate our work with the following experiment and explanation.
>
> ---
> **Q1:** What if the adversary only uses the part of the API response
>
> **A1:** We add a new experiment with the attacker trained on mixed API responses for the translation task, with X% of data watermarked, and (1-X)% of data not watermarked.
>
> | Watermark Percentage (X%) | p-value    | BLUE |
> | :---        | :----:  |   :----:   |
> |20| 31.1 | > $10^{-1}$ |
> |40| 31.1 | > $10^{-1}$ |
> |60| 31.0 | <  $10^{-4}$ |
> |80| 31.0 | < $10^{-7}$ |
> |100| 30.8 | < $10^{-7}$ |
>
> As shown in the table above, CATER is effective when more than 60% of data is watermarked. This means that at least 40% of data should not be from the (cheap) SOTA API. If looking for human annotation, it would cost $1.3M for 1M samples in our translation experiments according to the cost estimation in [1].
>
> We hope our experiments have consolidated our work.
>
> ---
> **Q2:** Elaborating on Figure 5 and its interpretation.
>
> **A2:**  As described in **Watermarking Algorithm Leakage**, we assume that attackers know the dictionary and conditions we used but not the exact watermarks. Then they can use this knowledge to find a list of suspicious groups (or entries). Each group meets the rules we used: 1) all words in this group are semantically equivalent, 2) except for one word, the occurrences of other words are zeros in the watermarked corpus. The number of these groups of each condition (or order) is the y-axis of the orange line. For the upper bound, as described in **section 3.3**, the total number of possible entries is  $|\mathcal{C}|=|\mathcal{F}|^K$ times the number of words.
>
> We hope we have clarified the statements for Figure 5 and section 4.2.
>
> ---
> **Q3:** The authors only study CATER for the English-centric datasets. Probably, the authors could extend CATER to other languages in the future.
>
> **A3:** Thank you for the suggestion. As a pioneer work, we investigate the effectiveness of our approach on English datasets. We are optimistic about adapting our approach to other languages because all the linguistic features used in this work, i.e., POS tags [2] and dependency tree [3], could easily be acquired for many other languages, such as German, French, etc.
>
> ---
> [1] Beyond Model Extraction: Imitation Attack for Black-Box NLP APIs. Xu el al. 2021
>
> [2] FLAIR: An Easy-to-Use Framework for State-of-the-Art NLP (Akbik et al., NAACL 2019)
>
> [3] UDapter: Language Adaptation for Truly Universal Dependency Parsing (Üstün et al., EMNLP 2020)

---

> > ### Comment · Reviewer_hUhu · 2022-08-09
> > **Thanks for authors' response**
> >
> > I would thank the authors for their detailed response. I choose to maintain my score.

---

> > > ### Author Response · Authors · 2022-08-10
> > > **Thanks to Reviewer hUhu**
> > >
> > > We would like to appreciate the reviewer’s encouraging comments and positive feedback, which has helped us polish our submission.

---

### Official Review · Reviewer_EoSA · 2022-07-11

**Rating:** 6
**Confidence:** 3
**Soundness:** 3 good
**Presentation:** 2 fair
**Contribution:** 2 fair

**Summary:**


The paper proposes a new watermarking framework for text generation APIs. Compared with the existing framework, the new framework increases stealth by minimizing the distortion of overall word distributions and incorporating high-order linguistic features. The experiments show that the new framework can identify imitation models with less capability than the existing method and keep the utility. However, it is proven to tolerate adaptive attacks, so it can be applied to protect text generation APIs better.

**Questions:**

Comment on the availability of practical attacks toward [10].

In section 4.2, you state that "malicious users would have difficulty in removing watermarks from the responses; unless they lean toward modifying all potential watermarks.". According to the article provided below, it seems to be possible to modify the watermarks without much loss of utility. Is that a threat to your framework? Please, clarify.

[A] Cracking White-box DNN Watermarks via Invariant Neuron Transforms. https://arxiv.org/abs/2205.00199


**Strengths And Weaknesses:**

pros:

- an important research problem.
- the proposed approach is carefully justified from both theoretical and empirical perspectives.
- high-quality paper writing.

Cons:

- Research motivation is questionable. Is there any concrete attack toward [B]?
- lack of evaluation under active attackers. Can you justify the strength of your defense under [A]?
- Novelty comparing to [B] is incremental. You achieved nearly the same performance with [B] in Table 1. Moreover, some (broken) texts in the current manuscript are from [B]. The authors should take a careful pass to paraphrase the paper.

[A]: Killing One Bird with Two Stones: Model Extraction and Attribute Inference Attacks against BERT-based APIs. https://arxiv.org/abs/2105.10909

[B]: Protecting Intellectual Property of Language Generation APIs with Lexical Watermark. AAAI 2022

The new framework is proposed to improve the stealth of the existing method. They are different in the optimization method and linguistic features for conditions. The optimization method is creative. The design for linguistic features is referred to another existing method.

In terms of the quality, the paper provides sufficient theoretical analysis to prove the feasibility of the new framework. In addition, there are empirical experiments conducted to verify the ability of the new framework. However, there is no direct comparison between the stealthiness of the new framework and the existing framework. Please justify if that was impossible or not needed.

Technical novelty and empirical results comparing to [B] seems incremental, particularly from Table 1. And the research motivation is also questionable. The improvement is based on the defense of a specific method of attack. However, there is no concrete breach of such an attack. The paper also lacks a deep description of this attack method, including the process of distorting the result of identification and the performance of attacking. Without concrete attack information, the application of the new method seems to be less urgent, in its current form.

---

> ### Author Response · Authors · 2022-08-02
> **Response to Reviewer EoSA**
>
> Thanks for your valuable and constructive feedback. We hope the following clarifications will address your concerns.
>
> ---
> **Q1:** Research motivation is questionable and technical novelty.
>
> **A1:** We respectfully argue that our work is well-motivated and novel to the research of watermark for NLG.
>
> Firstly, with regard to *research motivation*:
> 1. We observed and unveiled the drawback of the existing state-of-the-art NLG watermarking approach, which is vital to the research of security. Specifically, one can identify the watermarked words by comparing the word distributions of watermarked corpus and external benign corpus. Consequently, real-world attackers will endeavor to remove the watermarks to be exempt from the watermark verification. Our work should not be undervalued merely because *we prevent problems before they happen*.
> 2. Our research not only exposes risk but also proposes a defense solution. We hope the motivation for defending technology is also appreciated.
>
> Secondly, with regard to *technical novelty*:
>
> 1. *[Optimization Technology]:* Different from the heuristic approaches by [B], we propose to cast the watermarking process into an optimization problem. Solving this optimization problem guarantees that our approach can watermark imitation models and are robust to various adaptive attacks empirically and theoretically. Such conversion and the corresponding optimization method are novel, as acknowledged by the other three reviewers.
> 2. *[Linguistic Feature]:* The usage of linguistic features in our work is totally different from those used in [B]. [B] utilized **synonym** and **British/American spelling** for deciding *the word substitution sets*, while we use compositional **POS** and **DEP** features for deciding *the condition of substitution*. We consider both technology and purpose to be different to [B].
> 3. *[Performance]:* Our approach achieves on-par performance with [B]. Moreover, the new watermarking approach is more invidible and resilient to different adaptive attacks, even though the attackers have held strong prior knowledge about the details of our defense, as shown in Section 4.2. In addition, we also provide theoretical proof in Section 3.3 to consolidate the argument of the robustness of our approach.
>
> Finally, with regard to *research scope*: Our work is not simply proposing a watermarking approach, which is more invisible than [B]. Our method is provably to be able to expand the extremely imbalanced condition space to infinite. Empirically, it achieves inspiring results using finite order of linguistic feature combination.
>
> ---
> **Q2:** Justifying the strength of our defense under [A].
>
> **A2:** Thanks for the pointer. To the best of our knowledge, [A] aims to infer private attributes of the inputs of classification tasks, and the private attributes are irrelevant to the classification labels. However, this setting is not feasible to directly apply to text generation tasks, as all input information tends to be task-relevant. Specifically, for machine translation, the translation model must translate all information from the source language to the target language.
>
> ---
> **Q3:** Direct comparison to [B] in terms of stealthiness.
>
> **A3:** From our point of view, the stealthiness is decided by whether the watermark can be detected by attackers. Given such a definition, we have analyzed the stealthiness of [B]. Figure 1 shows that attackers could identify watermarks by analyzing the word distribution. In contrast, our approach is motivated and designed to minimize the word distribution shift. We have added a new figure to demonstrate the ratio change of word frequency of top 100 words between benign and watermarked (CATER version) corpora in the revised version (*see Figure 7 in our modified draft*). All conditional watermarked words are robust to word frequency attack.
>
> Moreover, we would like to highlight another property of our conditional watermark method. Because it selects a fraction of words for substitution, the total number of the watermarked words should be less than the number in [B]. Therefore, i) we cannot compare the p-value directly with the methods with more test samples, but our method achieves a decent p-value which is enough for verifying API ownership (see p-value in Table 1); ii) less watermarked words mean less harm to the semantic of the original outputs (see BLEU/ROUGE/BERTScore in Table 1).

---

> > ### Author Response · Authors · 2022-08-02
> > **Response to Reviewer EoSA (Cont.)**
> >
> > **Q4:** The details of cracking watermarks used by [B].
> >
> > **A4:**  After collecting the watermarked corpus from the victim model, we randomly select 5M sentences from common crawl data, which do not overlap with the training data of the victim model. We denote this dataset as the benign corpus. Then we obtain the word distribution for the watermarked and benign corpora, respectively, denoted as $P_w$ and $P_b$. Next, we take the union of top 100 words of both watermarked and benign corpora to obtain the suspicious words $S$. Now, we can calculate the ratio change of word frequency of each word in $S$ and plot them in Figure 1. It is clear that watermark words have the most significant ratio change. We have added the description to Appendix E in our modified draft.
> >
> > We hope the above description explains our word frequency analysis and solves your concerns.
> >
> > ----
> > **Q5:** According to the article provided below, it seems to be possible to modify the watermarks without much loss of utility. “Cracking White-box DNN Watermarks via Invariant Neuron Transforms.”
> >
> > **A5**: We would like to thank the reviewer for providing this pointer. Cracking all possible watermarks is not a wise choice for attackers, given the conditions are invisible to them, because they would probably have to modify all possible watermark words in the substitution set (in total 200 words with high frequency). There are several differences making it implausible to be adapted to our work:
> > 1. They are under the white-box setting, while NLG APIs are black-box to attackers;
> > 2. Their approach works on continuous space, while our watermarks conduct on discrete signals, i.e. word sequences. In other words, our watermarks become like task-relevant patterns, which force the imitation models to learn them. If the models are unable to reproduce the watermarks, then they cannot conduct the generation task properly.
> >
> > Although this paper was available on ArXiv in May, parallel to our submission, we are happy to incorporate the corresponding discussion into our work.
> >
> > ---
> > [A]: Killing One Bird with Two Stones: Model Extraction and Attribute Inference Attacks against BERT-based APIs. https://arxiv.org/abs/2105.10909
> >
> > [B]: Protecting Intellectual Property of Language Generation APIs with Lexical Watermark. AAAI 2022

---

> > > ### Author Response · Authors · 2022-08-07
> > > **Thanks to Reviewer EoSA**
> > >
> > > Thanks again for your valuable detailed comments.
> > >
> > > We have clarified the research motivation and technique novelty of our work in the previous response. In addition, following your suggestion, we have compared our approach with the previous work in terms of stealthiness. Finally, we discussed the feasibility of cracking our approach using a strong watermark removal technique for white-box models.
> > >
> > > We are more than happy to respond to any further questions if you still have concerns. We truly appreciate this opportunity to improve our work and shall be most grateful for any feedback.

---

> > ### Comment · Area_Chair_xKXG · 2022-08-09
> > **rebuttal response**
> >
> > Given your lack of response to the authors, despite their best efforts to engage you, I am not sure how to interpret your stance.
> > Could you please clarify your position on the paper? Thank you, AC

---

> ### Comment · Reviewer_EoSA · 2022-08-10
> **Thanks for the clarification**
>
> Review EoSA here. Thanks a lot for clarifying my questions and confusions (particularly on the related works) about this paper. I also quickly went through other reviews and your rebuttal. Given my relatively limited expertise in this field (comparing to other reviewers), I would like to raise my score to 6.

---

> > ### Author Response · Authors · 2022-08-10
> > **Thanks to Reviewer EoSA**
> >
> > We would like to appreciate the reviewer’s invaluable feedback, which has helped us improve our submission.

---

### Meta-Review · Area_Chair_xKXG · 2022-08-24

**Recommendation:** Accept
**Confidence:** Less certain

**Metareview:**

The authors propose a watermarking technique (CATER) to claim ownership of text generation APIs in the presence of imitation attacks. Their main idea is based on the observation that in the state of the art by analyzing the word frequency in API responses as well as publicly available data, an adversary's odds to learn the watermark increases. To remedy this, CATER conditionally watermarks the response to prevent the adversary from deciphering the watermarking keys.

Reviewers found the topic of the paper timely, is writing clear, and the overall contribution sound and of interest to the community.

**Award:**

No

---

### Decision · Program_Chairs · 2022-09-14

Accept